# A Machine Learning-Based Method for Automated Blockchain Transaction Signing Including Personalized Anomaly Detection

**DOI:** 10.3390/s20010147

**Published:** 2019-12-25

**Authors:** Blaž Podgorelec, Muhamed Turkanović, Sašo Karakatič

**Affiliations:** Faculty of Electrical Engineering and Computer Science, University of Maribor, Koroška cesta 46, 2000 Maribor, Slovenia; muhamed.turkanovic@um.si (M.T.); saso.karakatic@um.si (S.K.)

**Keywords:** blockchain, transactions, digital identity management, anomaly detection, machine learning

## Abstract

The basis of blockchain-related data, stored in distributed ledgers, are digitally signed transactions. Data can be stored on the blockchain ledger only after a digital signing process is performed by a user with a blockchain-based digital identity. However, this process is time-consuming and not user-friendly, which is one of the reasons blockchain technology is not fully accepted. In this paper, we propose a machine learning-based method, which introduces automated signing of blockchain transactions, while including also a personalized identification of anomalous transactions. In order to evaluate the proposed method, an experiment and analysis were performed on data from the Ethereum public main network. The analysis shows promising results and paves the road for a possible future integration of such a method in dedicated digital signing software for blockchain transactions.

## 1. Introduction

One of the main innovations of blockchain technology is the ability for users to transfer the ownership of digital resources (e.g., cryptocurrency) directly to each other within a decentralized and distributed peer-to-peer network, without the need of third party (e.g., financial institution). The execution of transactions transfers the ownership of resources, and the global state stored in the blockchain network is modified. Each transaction broadcasted to the blockchain network must be digitally signed by the user, performed by a digital signature algorithm. The usage of digital signatures as one of the main building blocks of blockchain technology is necessary to guarantee the integrity and the non-repudiation of the transactions [1,2].

Concerning the results of studies [3,4,5,6,7] on the usability of the digital signatures, there are still many obstacles preventing users from accepting the usage of the digital signatures within their everyday usage. These obstacles are related to managing, controlling, and indeed using cryptographic keys. The studies conclude that such tasks are complex and consequently overwhelming for users. Moreover, studies have pointed out that these usability issues also have an impact on users’ security, e.g., users are unable to recognize potential intrusions while being involved in a digital signing process. The work of [8], while analyzing the issue, concludes that similar obstacles still exist.

Limited to the most mature blockchain platform (i.e., Ethereum, see Available online: https://github.com/ethereum (accessed on 7 November 2019)), a user’s blockchain-related digital signature is associated with their blockchain account (i.e., public-private key). The signing process is, however, performed within dedicated software—so-called (digital or crypto) wallets. With a wallet, users control the digital signing process and perform interactions in the blockchain network, including decentralized applications. Each blockchain-related account has a unique address—a hashed version of a public key, representing an address of a user’s digital identity in the blockchain network. While transferring digital resources (e.g., cryptocurrencies) over the blockchain network using cryptographically signed transactions, the addresses play a crucial role in defining the new owner of the resource. Since every blockchain transaction needs to be digitally signed by the user manually, this can lead to security issues.

Most of the aforementioned digital signatures’ usability issues are related to the complexity of the digital signing process. In relation to the blockchain technology, this can affect the security of the digital assets owned by users because they are eligible to transfer digital assets only with a correctly and successfully digitally signed transactions. Each time a user wants to transfer a digital asset, a digitally signed transaction needs to be provided, which can be performed only within the aforementioned complex process of digital signing. In the case of frequent transfers over a longer time period, the digitally signing process can be performed superficially, thereby affecting the security of the digital assets. In this situation, potential malicious counterparties can use such user behavior as an advantage and attempt to propose to users the signing of transactions that have adverse impacts on their digital resources. For example, the user performs a transaction, through a web application (i.e., decentralized application), in which the goal is to transfer some amount of cryptocurrency. Within this process, the web application prepares and presents the transaction to the user for the digital signing process. After digitally signing the transaction, it is broadcasted to the blockchain network and irreversibly executed. A problem may occur if the application generates a malicious transaction, with tempered data, e.g., a higher amount than initially specified by the user. The user could sign the fabricated transaction, which later cannot be undone, consequently resulting in a loss of funds for the user. However, this paper proposes a machine learning-based method that simplifies the digital signing process. The method provides automated digital signing, while also using anomalous transaction detection to ensure the prevention of the aforementioned attack scenario.

The objective of this article is to propose a novel method that will enable an automatic digitally signing of blockchain transactions, thus relieving the user of signing and reviewing each and every transaction. The proposed method provides an automatic and personalized digital signing process of blockchain-related transactions, including additional security features, i.e., an anomaly detection mechanism, which is based on the personalized transaction data of the user. By incorporating the proposed method into blockchain dedicated software (e.g., wallet) for managing the digital signature process, the designated transaction, which includes the transfer of cryptocurrency from address A (i.e., sender) to address B (i.e., receiver), is automatically digitally signed on behalf of the sender, except in the case of a potentially anomalous transaction detection, which can potentially produce damage to the sender. In such a scenario, the manual approval from the sender to digitally sign such a transaction is required. Properties of the proposed method also help to circumvent existing digital signatures usability issues that can be indirectly present in the usage and, with this, constitute an obstacle to the wide adoption of blockchain technology.

The contribution of this paper can be summarized as:a novel machine learning-based method for automated digital signing of blockchain transactions, which includesa personalized anomalous transaction detection system, andoperates and stores the personalized data from the anomaly detection model on the user’s local environment, instead of a 3rd party centralized environment.

The rest of the paper is structured as follows. The next section reviews the related work on detecting anomalous transactions in blockchain technologies. The third section presents the current state of the signing of the transactions in blockchain technologies. The fourth section presents the proposed method of the automated and personalized signing of blockchain transactions in detail—from the transaction data pre-processing, the whole procedure of the proposed method, and comparison to the current state. Section 5 presents the results of an experiment—first on the detection of the anomalous transactions, and next on the differences in transaction patterns of individual addresses. The paper concludes with final remarks and the plans for future research on the subject.

## 2. Related Works

Related works in the field of fraud detection on the blockchain network have been reviewed. Pham and Lee in [9] consider the general problem detection of suspicious users and transactions on two graphs generated from the Bitcoin blockchain network. On the first graph, nodes are represented as users and, on the second, as transactions. For the detection of anomalous behavior, they have used k-means clustering, Mahalanobis distance, and Support Vector Machine (SVM) machine learning methods.

In [10], authors have extended the same research by using Power Degree & Densification Laws, along with Local Outlier Factor unsupervised learning methods, within which they achieve similar results as in [9].

In [11], they applied and compared the capabilities of Random Forests, Support Vector Machines, and XGBoost supervised learning methods to detect fraudulent accounts in the Ethereum blockchain network.

Another work [12] proposed an approach with trimmed k-means unsupervised learning method, which is capable of simultaneous object clustering, for the anomaly detection in the Bitcoin network. The evaluation was performed with the same data set as used in [9,10]. The proposed approach gained better results with more fraudulent detected transactions.

To the best of our knowledge, there are no existing solutions whose purpose is to introduce machine learning methods into the blockchain transaction digital signing process, intending to enable automatic digital signing of transactions, while additionally actively detecting potentially fraudulent transactions which are subject to the digital signing performed by the user. Although the work on detecting anomalies among the transactions has already been done, none of the reviewed methods adapts to the address’ transaction patterns, and none reviewed the approach of using user data which are labeled (the anomalies are not known). The novelty of the proposed approach is that it can be used with any address that has a sufficient history of the transaction, regardless of the transaction patterns, and identified anomalous transactions.

## 3. Blockchain Transactions

This section gives a brief overview of blockchain transactions, usage of cryptographic digital signatures, and the signing process of the transactions itself. While discussing the topics above, we will focus on the Ethereum platform, as a platform with the most number of active developers and researchers [13] and also used in the experimental part of this research. The Ethereum platform further allows the building of more complex applications (i.e., decentralized applications), where code that implements arbitrary rules (i.e., smart contracts) can directly control digital assets (e.g., cryptocurrency, coins, smart properties) [2]. In the blockchain network, transactions are in a secure and immutable manner sealed in blocks, which are chained together into a ledger [14]. Each new block itself contains several transactions verified by digital signatures [1], which are recorded as a state in a distributed ledger shared among all participants in the blockchain network. Consequently, this provides transparency and the immutability of transactions performed within the network [15]. The state in Ethereum is made up of the accounts, where each account has a unique 20-byte address, and there are two types of accounts. Users with their private keys control Externally Owned Accounts (EOA), while code implemented in smart contracts control Contract Accounts (CA). The state can be changed through the transactions performed and sent to the blockchain network by the EOA or CA, while a transaction recipient can be an EOA address (i.e., smart contract) or CA address (i.e., user) [2,16].

Transactions relate to a digitally signed data package that includes a message and can be sent only from an EOA. Besides the already mentioned recipient, Ethereum transactions also contain a digital signature, which identifies the transaction sender, the amount of native cryptocurrency Ether that will be transferred from the sender to recipient, and an optional data field, which can be used in the case that the recipient is an EOA. In order to prevent denial of service attacks on the Ethereum blockchain network, each transaction must define a limit that denotes how many computational steps the transaction execution transaction is allowed to take, expressed in native cryptocurrency Ether. The more computationally expensive the transaction, or the more the amount of data stored in a state as part of the transaction is increased, in turn, increases the steps required for a transaction to be successfully performed. Besides this, the transaction also needs to include a value representing the fee, which the sender is willing to pay per each computational step [2,16]. Users sign and publish transactions using dedicated software, or wallets. There are numerous types of wallets, e.g., paper wallet, mobile wallet, online wallet, hardware wallet, and a desktop wallet. The common feature of each wallet type is that they all handle a user’s private key, which is required for performing actions (i.e., transactions) within decentralized applications [3,17].

### 3.1. Current State of the Digital Signing Process

In this article, we limit ourselves to the transactions, which recipients define as either CA or EOA and are broadcasted to the blockchain network through the decentralized application, i.e., an application which leverages on the blockchain [18]. A user interacts with a decentralized application and performs digital signing of transactions with a desktop wallet, which keeps a user’s private key on the user’s local machine, encrypted with a password. To describe the current process of user interaction with a decentralized application and, consequently, with a blockchain network, we define the following use case scenario: a user with a desktop wallet wants to transfer some amount of cryptocurrency (e.g., Ether) through a decentralized application to another CA or EOA address. The described use case scenario is presented in Figure 1.

To use the wallet, a user has to unlock the wallet with the proper password. With this, he gets access to his blockchain account (i.e., private key) and a possibility for establishing a connection to the blockchain network (e.g., Ethereum public main network). With an unlocked wallet and established connection to the blockchain network, he gets the ability for the usage of decentralized applications. Later, the user opens an interface of a decentralized application in a browser, which consists of a form where the user needs to fill an amount of cryptocurrency to be transferred and a recipient address. After the form is filled, the user confirms such an action with the click on the button. The decentralized application prepares a transaction, which is sent back to the user to be manually reviewed and digitally signed. At this point, a manual review from the user is required. Manual review is necessary for the purpose of users themselves deciding if the transaction proposed by the decentralized application is valid (i.e., is not fraudulent) and does not cause any potential unwanted loss of funds (i.e., cryptocurrency) to the user. The user has the option to confirm a transaction, i.e., digitally sign a transaction, that approves a transfer of cryptocurrency to the recipient or rejects a transaction that interrupts the transfer of cryptocurrency to the recipient. If the user chooses to sign a transaction, this transaction is sent to a decentralized application and broadcasted to the blockchain network. After the blockchain network processes the transaction, the transfer is successfully performed, and its results are irreversible.

## 4. Proposed Machine Learning Method for Automated Digital Signing

In this section, we present the proposed method for automated and personalized digital signing of blockchain transactions. We also present the background of personalized anomaly detection for blockchain transactions, as well as the personalized anomalous blockchain transaction detection system. Furthermore, we will demonstrate how our proposed method enables an automated review and digital signing of transactions proposed by decentralized applications instead of the current manual approach, which will be applied to the identical use-case, as specified earlier.

### 4.1. Personalized Anomaly Detection for Transactions

The history of one address’ transaction is in the form of time series data, where data is stored in sequence along the timeline. Much research has been done on the detection anomalies on time series data [19,20,21,22], reviewed in detail in [23,24]. Traditional time series analysis methods only deal with the variation of the observed data and not with its frequency and amount combined [21]. Another approach in analyzing a series is the usage of neural networks—more specifically, recurrent artificial neural networks (RNN), with the usage of Long Short-Term memory architecture [25,26] and its variations. The drawback in such RNN approaches is that the training of the model requires a lot of data, which consequently takes a lot of time. As the transaction data of one address are limited, there is a limited possibility to construct the anomaly detection system using RNN for time series. Hence, the traditional anomaly detection methods [27,28,29], which work with a small amount of data and take less time to build the model, should be used. The challenge with traditional machine learning anomaly detection methods is that they are not suited for time series analysis due to the sequential nature of time series data. Consequently, there is a need to extract the features from the series data to be used in the traditional anomaly detection methods. Numerous approaches have been proposed and used in the literature [30,31], which inspired our proposed approach and are described in the following.

### 4.2. Transaction Processing

Each address has a list of historical transactions, from which we use the following data: the timestamp of the transaction and the transaction value in its corresponding USD value at the time of the transaction. As we are dealing with time series analysis, we utilized the method of rolling window aggregation [32] from which the new features were extracted. The rolling window feature extraction for time series is a procedure, where time-series data are analyzed in sequence, from the earliest data in steps of size *h*, which could be defined as the number of measurements in the sequence or as the time frame (if the series has the time defined). The size of the rolling window is defined as *w*, which is the number of measurements in one window. The basic procedure of feature extraction for the last transaction, denoted by the gray color of the background, is shown in Figure 2.

Each of the time windows forms a new sequence, which is then used to calculate the aggregations of the sequential data. In this research, the size of the rolling window *m* was defined as the time frame and was not fixed to the number of individual transactions. In addition, the step size *h* was defined as one transaction. Since any blockchain address could have different patterns of transactions, and the goal of our research was to make a personalized anomalous transaction detection, several different time frames as windows sizes *m* were used: *one second*, *one minute*, *one hour*, *one day*, *seven days*, *14 days*, *30 days*, *60 days*, and *90 days*. Consequently, as the number of transactions varies across the windows, the size of the rolling window is not fixed. The goal of our research is not to determine which of these time frames is most relevant in general, but to build a procedure, which determines the appropriate time frame for each of the addresses individually.

The aggregation functions *f* used on the transaction windows were the following statistical procedures: *mean*, *median*, *standard deviation*, *sum*, and *count* (number of transactions) of the transactions in the window. Aggregating the number of transactions enables the transformation of the time series data to tabular data (shown in Figure 2) and the usage of traditional anomaly detection methods. The combinations of time frames *w* and aggregation functions *f* are denoted as the combination $w_{i}f_{j}$, for each i∈[1,v] and j∈[1,k]. Value *v* is the number of different time frames used for windows sizes (in our experiment 9), and value *k* is the number of different aggregation functions used on the set of transactions in the windows (in our case 5). The extracted features for each transaction were thus the combinations $w_{i}f_{j}$ of all of the time frames *w* and all of the functions *f*, resulting in 45 new extracted features in addition to the original USD value of the transaction.

### 4.3. Proposed Personalized Anomalous Transaction Detection System

Before the creation of the anomaly detection method for one particular address, there needs to be enough data available to reveal the patterns of the transaction of that particular address. Here, we set the number of the minimum needed transactions to 100, which was experimentally determined to be enough to reveal basic transaction patterns already. In future work, this is one of the parameters which could use future focus but is out of the scope of this research. After the first 100 transactions, the feature extraction process, described in the previous section, starts, which converts the time series data to tabular data. Then, the anomaly detection method is used to construct the anomaly detection model. For the next *j* transaction, this model is used to evaluate each transaction, to be either normal or anomalous. Again, the parameter *j* for the number of transactions evaluated by the same anomaly detection model can be further optimized to the individual address but is again out of the scope of this research, where we tried to evaluate the usefulness of the proposed approach. Algorithm 1 shows the pseudocode of the procedure.

As our objective was to test the validity of the proposed method and not to find optimal settings, we limited ourselves to using only one unsupervised anomaly detection method—the Isolation Forest for novelty detection method by [33], which isolates each transaction and splits them into inliers (i.e., normal transaction) and outliers (i.e., anomalous transaction), based on the number of decisions in the decision tree to isolate the transaction. The nature of anomaly detection problems without labeled cases is such that there is no ground truth for the addresses. In other words, there is no ability to determine if the anomaly detection method has correctly labeled the transaction as anomalous or not. This is sometimes called novelty detection. Hence, we presented the results of unsupervised anomaly transaction detection in a time series chart, where detected anomalous transactions are clearly denoted.
**Algorithm 1:** Pseudocode of proposed method for identification of anomalous blockchain transactions.
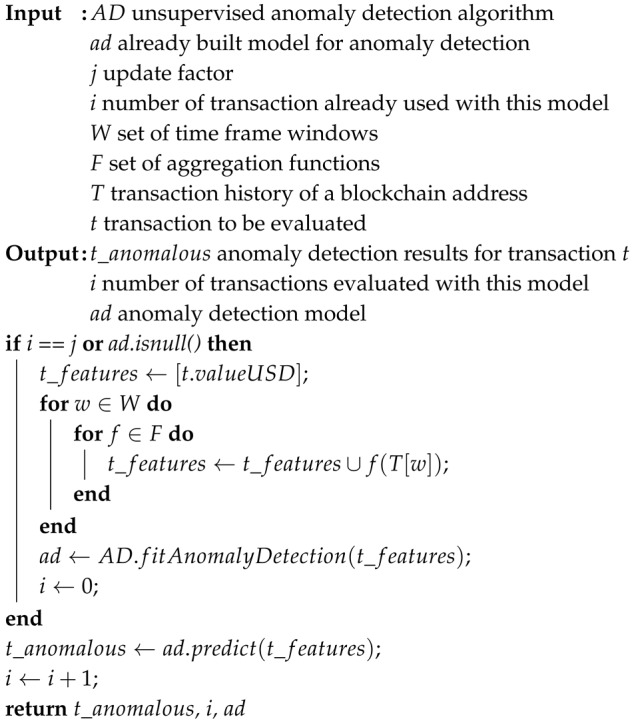


### 4.4. Digital Signing Using the Proposed Method

Figure 3 presents the same use-case scenario as defined in Section 3.1. However, unlike before, the transfer of some amount of cryptocurrency through a decentralized application is supported by the system, which includes our proposed method for automated digitally signing of blockchain transactions. Similarly, as in Section 3.1, a user first has to unlock a wallet and establish a connection to the blockchain network to get the ability for the usage of decentralized applications. Through the form on the interface of a decentralized application, the user inputs information about the amount of cryptocurrency to be transferred and a recipient address. After the user sends a form, as in Section 3.1, the decentralized application prepares a transaction that needs to be signed by the user to finalize the transfer of funds. Unlike the current process depicted in Section 3.1, where the user manually reviews the transaction, the transaction is evaluated by the add-on system, which involves our proposed method for detecting anomaly blockchain transactions. Furthermore, in the case a transaction is marked as valid, based on past user behavior (i.e., performed transactions), it can be automatically signed and send back to the decentralized application without requiring any additional actions from the user. If the system evaluates a transaction as fraudulent, a transaction is sent to the user for manual review. Such a transaction can still be digitally signed and broadcasted to the blockchain network through the decentralized application, which can only be done manually by the user.

## 5. Experiments with Proposed Method

We tested the proposed anomalous transaction detection method on 10 Ethereum addresses, of which properties are presented in Table A1 in the Appendix A. The transactions of the addresses were on 20 September 2019, gathered from the Ethereum public main network. Starting from the block Nr. 8578851, we evaluated every block to find enough unique addresses to satisfy the following requirements. The address that appeared in a block must be controlled by private key (i.e., CA), have at least 2000 but no more than 6000 outgoing transactions (i.e., digitally signed) that include a transfer of any amount of cryptocurrency (i.e., Ether), and must be performed within a period no longer than three months.

Other settings of the methods used in the experiment were the following. Isolation Forest consisted of 100 individual decision trees with the contamination factor of 0.01. The Random Forest classification algorithm was used to determine the feature importances and was also constructed out of 100 individual decision trees, built with Gini impurity split criteria. The experiment was implemented using the scikit-learn v0.21.3 Python package. All of the other settings were left at their default values.

### 5.1. Results

As is normal for unsupervised anomaly detection, there is no precise way to determine if the transaction was a mistake, an error, or a fraudulent transaction, without contacting the address owners. Without the ground-truth labels, there is no clear way to evaluate the efficiency of the proposed method. For this reason, all of the results are presented in visual form for the readers to interpret and evaluate the proposed approach. The results of anomalous transaction detection are presented in a time series chart form in Figure 4. Anomalous transactions are denoted as red crosses, and normal transactions are denoted as blue dots. The first 100 transactions were only used as the starting training data, thus separating them with the dashed line.

The chart in Figure 4 shows that the proposed method labeled the transaction as anomalous when either the amounts of the transaction were out of the ordinary (unusual high transaction values) or there was an unusually high frequency of transactions at one point in time. The unusually high frequency of transactions is especially evident in the top left sub-plot for the address starting with 0xFC04..., where there is a clear peak in the frequency of transactions from May through the end of July 2019. Note that there were also some of the transactions labeled as anomalous that bellowed the value trend of the transaction at that time. The low value of transactions was not the only indicator of unusual activity; this was supplemented with an unusual count of the transactions at that time.

Looking at the other addresses, there is a clear difference in patterns of transactions among them (inter-address difference) as is the difference of patterns in the addresses themselves along the time (intra-address difference). The inter-address difference is clearly represented in the different y-axis scales of the graphs, as one address has a common transaction in the values of thousands of USD (bottom right address starting with 0xFa45...), and others operate with much lower values in hundreds of USD (bottom left address starting with 0xCac7...). This difference is a clear indicator that each address has its own pattern of the transaction, be it in frequency or in their values. Thus, we hypothesize that there is also a difference in the time frames that describe common address transaction patterns, and these are important in determining if the transactions are anomalous or not.

### 5.2. Personalized Approach in Detecting Anomalous Transactions

To determine if the personalized approach is valid, we analyzed the important feature (transaction properties) for each individual address. This is done in order to test whether (1) there is a consensus among important transaction properties in identifying the anomalous ones or (2) they are specific to each address. If the features importances are consistent across all of the addresses, there is a clear pattern of transaction among all of the addresses; otherwise, the personalized approach is necessary.

The importance factors were calculated using the Random Forest classification algorithm [34], of which features were the same as the transaction data inputted into the Isolation Forest anomaly detection method, and the target values were the anomaly labels as outputted by the Isolation Forest. The resulting Random Forest model consisted of 100 individual trees, from which the feature importances were calculated, where the importance factor ranges in [0,1], in which the higher number denotes the higher importance. As we have several aggregations on every time window, all of the importances for one time window were summed to determine the importances of times for every address. The results are shown in the horizontal bar charts in Figure 5, where the length of the bar denotes the sum of all of the importance factors for that time frame; the results are also printed in Table 1, where they are supplemented with ranks.

The results from Figure 5 show that there is no consistent pattern of most important time frames among all 10 addresses in our experiment. The most important time frames vary from hour patterns (address starting with OxFC04...) to minutes (five addresses) and all the way to seconds (four addresses). The address starting with OxFC04... is also clearly different from the others, as the longer time frames are more important here than in other addresses, where time frames less than one day dominate the importance factors.

Table 1 of the ranks of importance factors for different time frames supports that there are no universally most important time frames in all of the ten analyzed Ethereum addresses. Here, each address was analyzed individually, and the time frames were ranked according to their importances. The results in Table 1 show the minimum, maximum, and mean ranks of these time frames, as well as standard deviation of the rank. According to this, the most important feature for the ten analyzed addresses is the *minute* because its mean rank is the smallest (1.8). This is followed by time frames second (mean rank of 2.0) and hour (mean rank of 3.3), and then the raw USD value of that individual transaction (mean rank of 4.2). The standard deviations of these ranks are relatively high (minimum of 0.9 for the hour time frame, and a maximum of 2.04 for one transaction), which shows that there is no clear consensus.

In this manner, we conclude that usage of a personalized approach for anomalous transaction detection methods is appropriate, as the inter-address differences have to address the anomalous transaction detection, which has to be specifically fitted to patterns to each address transaction activity individually.

### 5.3. Case Study

This part takes one address (starting with 0x491f...) to look into the intra-address difference in patterns. The top of Figure 6 shows the average USD value of transactions denoted with the blue line and the 95% confidence interval denoted as the blue shadow area. The narrower the confidence interval (from the second quarter of 2018 until the end of 2018), the greater the increase in (1) the similarity in the transactions’ values, (2) the width of the area (as first quarter of 2018 and first quarter of 2019), and (3) the scattering of the transaction values.

The bottom part of Figure 6 shows the number of transactions grouped by 30-day time frames. Both these metrics are grouped into 30-day intervals for the sake of clarity, despite the 30-day time frame not being the most important feature for this address. Even when looking at the mean values and number of transactions in the 30-day intervals, there is a clear intra-address difference. There is a clear higher number of high-value transactions at the end of the year 2017 and a drop of both in the first quarter of 2018. The values of transactions and the number of them is relatively consistent all the way to the first quarter of 2019, where there is a drop of both, with a small spike after that.

With these intra-address changes and the changing nature of the use of individual addresses in mind, the consistent refitting of the anomalous transaction detection was implemented. The length of the interval when the model has to be refitted is out of the scope of this paper, and it is a possible future research endeavor.

## 6. Conclusions

In this, paper we introduce a method for the personalized and automated signing of blockchain transactions. The presented approach employs an innovative usage of artificial intelligence within the blockchain technology domain, specifically machine learning methods, which are used to automate the signing process, while also securing the user against the digital signing of potential fraudulent blockchain transactions. The proposed method is envisioned to be included within the software that operates on top of blockchain technology and with so-called blockchain-based user wallets. Furthermore, the anomaly detection system, which is the core of the automated signing method, operates and stores the data (i.e., anomaly detection model) on a user device. As presented within the use-case defined in the paper, the method can be used to improve the usability of decentralized applications. Moreover, utilizing the method in the environments that require continuous performing of blockchain transactions (i.e., cryptocurrency exchanges) can prevent fraudulence or other malicious activities that may result in the loss of funds stored in wallets (i.e., hot wallets) that are used in regular exchange activities.

To evaluate the proposed method, we experimented with real-life transactions from the Ethereum public main network. Although there is no scientifically valid metric to determine the quality of unsupervised anomalous transaction detection, the proposed method returns promising results. Even if the transactions labeled as anomalous would not be frauds, users are only prompted to manually sign these unusual transactions. Transactions that are following the personal transaction patterns are still signed automatically. Manually signing only the unusual transactions still greatly improves the usability of the digital signing process and, consequently, contributes to more user-friendly usage of blockchain-based applications.

During the development of and experimenting with the proposed method, some interesting questions came along that could be further investigated in the future. As the user is prompted to sign an anomalous transaction, its answer could be used as the ground truth for the anomaly. This could be used as a reward for reinforcement learning or as a true label for the supervised learning method, which could replace the unsupervised approach. Next, the update factor *j*, which determines how long the model is used before it is refitted, could be one of the optimization factors and thus personalized to the patterns of the individual address. In addition, other unsupervised anomaly and novelty detection methods could be used in this approach—from statistical to nature-inspired meta-heuristic, as well as artificial neural networks with typologies optimized for small amounts of data. Finally, a performance comparison between the proposed method and the original process could be performed, which would also shed further light on the usability aspect of such an approach.

## Figures and Tables

**Figure 1 sensors-20-00147-f001:**
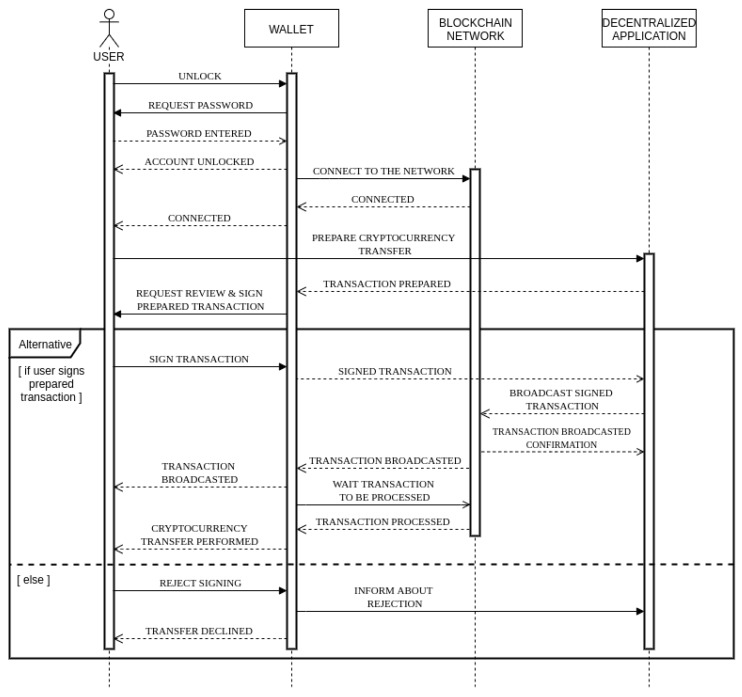
Current digital signing process using a decentralized application.

**Figure 2 sensors-20-00147-f002:**
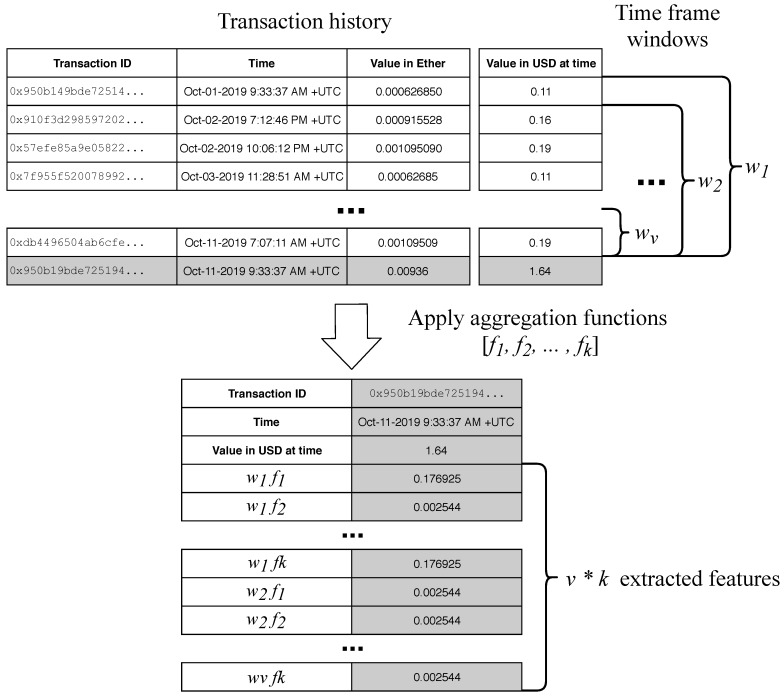
Current transaction processing procedure.

**Figure 3 sensors-20-00147-f003:**
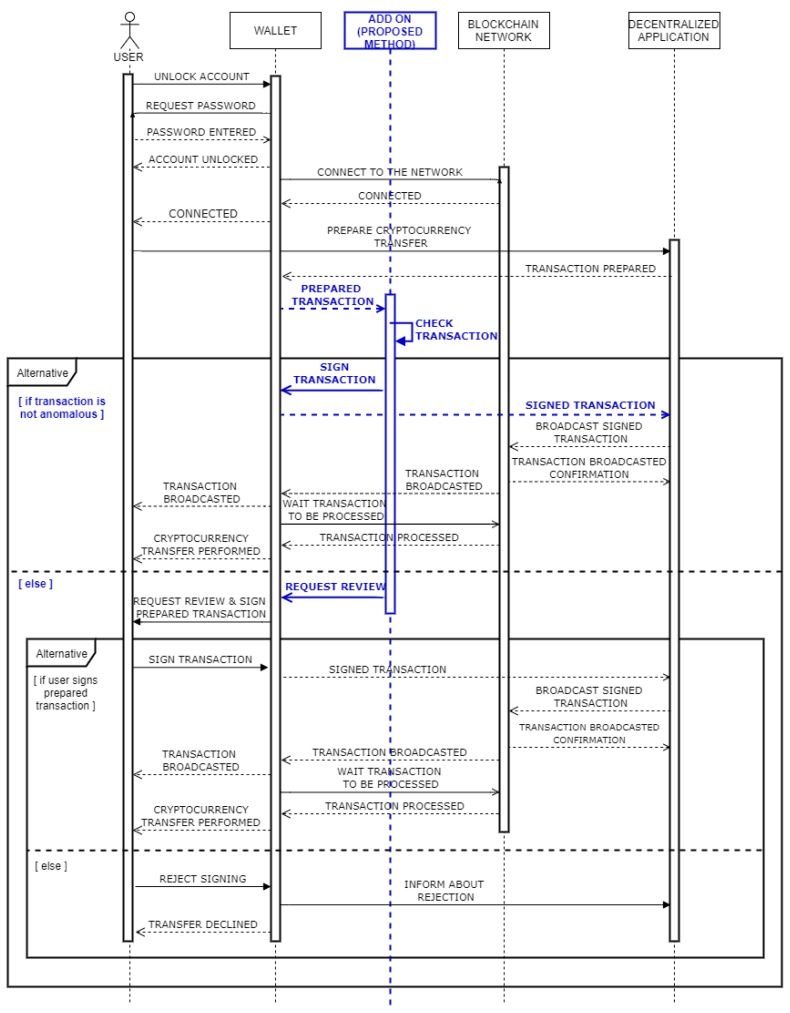
Transaction processing including the proposed method.

**Figure 4 sensors-20-00147-f004:**
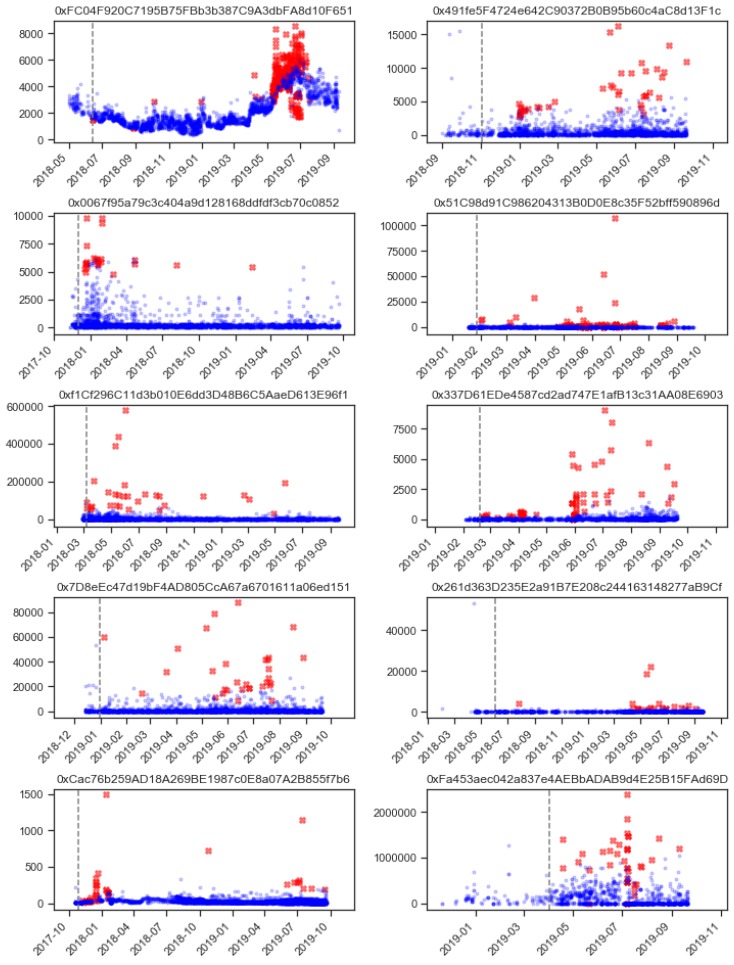
Transactions with annotated anomalies (red X) for all ten Ethereum addresses in the experiment.

**Figure 5 sensors-20-00147-f005:**
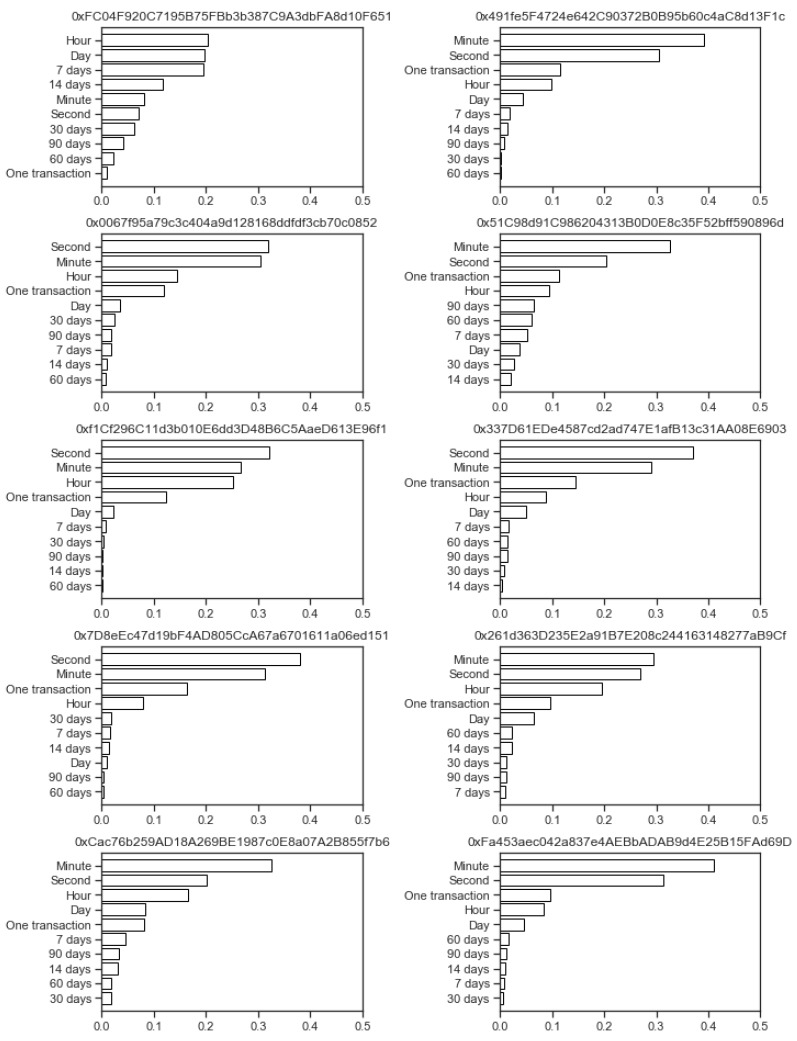
Important time frames for all ten addresses in the experiment.

**Figure 6 sensors-20-00147-f006:**
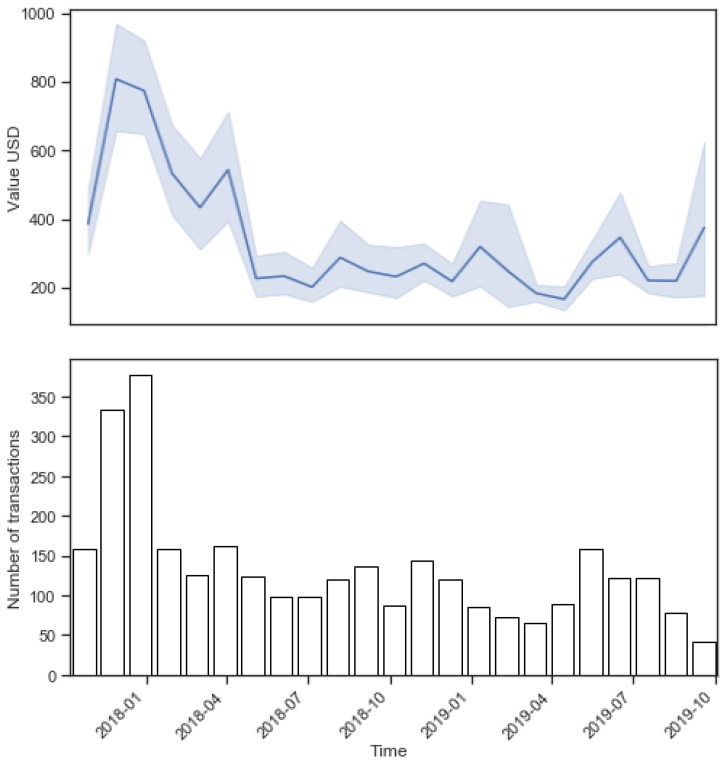
Sum of USD values (**top**) and number of transactions (**bottom**) for 30-day periods for one of the Ethereum addresses.

**Table 1 sensors-20-00147-t001:** Ranks for different time frames of feature extraction process.

Time Frame	Rank			
	Min	Max	Mean	Std. Dev.
One transaction	3	10	4.2	2.0400
Second	1	6	2.0	1.4140
Minute	1	5	1.8	1.1662
Hour	1	4	3.3	0.9000
Day	2	8	5.2	1.6610
7 days	3	10	6.7	1.8466
14 days	4	10	7.9	1.7000
30 days	5	10	8.0	1.6125
60 days	6	10	8.3	1.7349
90 days	5	9	7.6	1.1136

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
