# Peer review of "A Machine Learning-Based Method for Automated Blockchain Transaction Signing Including Personalized Anomaly Detection"

_sensors, 2019, doi:10.3390/s20010147_

Round 1

Reviewer 1 Report

Comments to the Authors

The authors proposed a novel anomalous transaction detection method using the Isolation Forest algorithm and improved the process of digital signing of blockchain transactions into an automated procedure. They also carried out a rigorous experiment using real-life transactions from the Ethereum public main network, which showed the proposed method returns promising results. However, there are still several points that need further attention.

1. The authors stated that 'there are still many obstacles preventing users from accepting the usage of the digital signatures within their everyday usage' in Line 22. However, these obstacles are not mentioned or explained below, which may bias the reader's understanding. Furthermore, 5 references (from no.3 to no. 7) are listed but not discussed in any way. I assume that the authors cited these articles to indicate 'these obstacles' exist. But only one of these articles (no. 3) is published in 2018, while others are published before 2006. I wonder if 'these obstacles' still bother the users today.
2. The introduction part could be improved by adding some description of an application scenario or a malicious attack scenario, which will make readers better understand why we need an anomalous transaction detection system.
3. The concept of 'single point of failure' has different specific meanings in different environments. The authors listed 'immune to a single point of failure' in the contribution part and the conclusion part, while the rest of the article does not explain or prove it.
4. Figure 3 is supposed to show the improvements of the proposed method compared to the original protocol. Highlight these improvements in a different color.
5. The proposed anomalous transaction detection method is implemented as an add-on to the wallet. Since the authors mentioned that the digital signing process is time-consuming, some experiments of performance comparison should be performed between the original method and the proposed method.
6. I appreciate the novelty of the article, but I am a bit worried if this article is out of the scope of Sensors. I hope the author could convince me. 

Author Response

First of all, we would like to thank both the reviewers for the important and helpful remarks, advice, and suggestions concerning the content of the paper. It helped us a lot since we noticed some aspects which had to be improved.

Reviewer #1:

-- The authors proposed a novel anomalous transaction detection method using the Isolation Forest algorithm and improved the process of digital signing of blockchain transactions into an automated procedure. They also carried out a rigorous experiment using real-life transactions from the Ethereum public main network, which showed the proposed method returns promising results. However, there are still several points that need further attention.

1) The authors stated that 'there are still many obstacles preventing users from accepting the usage of the digital signatures within their everyday usage' in Line 22. However, these obstacles are not mentioned or explained below, which may bias the reader's understanding. Furthermore, 5 references (from no.3 to no. 7) are listed but not discussed in any way. I assume that the authors cited these articles to indicate 'these obstacles' exist. But only one of these articles (no. 3) is published in 2018, while others are published before 2006. I wonder if 'these obstacles' still bother the users today.

We are sincerely thankful for pointing out the insufficiency. In order for the paper to be concise, we extended the introduction with a short explanation of the obstacles discussed in the cited references. Furthermore, we also added an extra reference, which confirms that the obstacles from the references mentioned above are still present and bother users today.

Changes in the revised version of the manuscript are visible at the following lines: 24-28.

2) The introduction part could be improved by adding some description of an application scenario or a malicious attack scenario, which will make readers better understand why we need an anomalous transaction detection system.

 We recognize that the application scenario, together with the malicious attack scenario, can help readers to understand better the need why to introduce an anomalous transaction detection system in the digital signing process, so we really appreciated the advice. For this purpose, we added a real-life application scenario, which describes, what are the results if a digital signing of the transaction is not performed carefully, and where and why the introduction of an anomalous transaction detection system can help the users.

Changes in the revised version of the manuscript are visible at the following lines: 48-57.

3) The concept of 'single point of failure' has different specific meanings in different environments. The authors listed 'immune to a single point of failure' in the contribution part and the conclusion part, while the rest of the article does not explain or prove it.

 We are sincerely grateful for this sighting and remark. We agree that the term 'single point of failure' was incorrectly used, which caused the agreeable confusion. Concerning this, we expanded and reformulated our contribution description, with more appropriate terms that match with the described concept regarding our proposed method. Due to the same referring problem, we have also correct the Conclusion section.

Changes in the revised version of the manuscript regarding this topic are visible at the following lines: 72-76 and 365-366.

4) Figure 3 is supposed to show the improvements of the proposed method compared to the original protocol. Highlight these improvements in a different color.

To additionally emphasize the improvements of the proposed method compared with the current process, as was recommended, we highlighted these improvements with a different color presented in Figure 3. Furthermore, we have also extended the process in the sequence diagram in order to present it comprehensively.

5) The proposed anomalous transaction detection method is implemented as an add-on to the wallet. Since the authors mentioned that the digital signing process is time-consuming, some experiments of performance comparison should be performed between the original method and the proposed method.

 We understand the concerns and reason why a performance comparison was raised up. However, during the research, the focus of it was directed towards the proposed model and evaluating it with real data from the public blockchain network. Even though we agree that the next logical step is a performance comparison, this however we think is a research uptake for itself, which requires detailed methodology and focus. One of the reasons for leaving it out in the current work, is the fact that, having a method, which automatically signs the transactions for the users, makes it trivially better in terms of time required from the user’s perspective. However, since the process would happen in the background, the users would not note this. Therefore it is crucial to construct a meaningful experiment, which we think should be a separate work.

 However, in order to include this thinking, we included this into the Conclusion section at the following lines: 388-390.

6) I appreciate the novelty of the article, but I am a bit worried if this article is out of the scope of Sensors. I hope the author could convince me.

 We are thankful and happy for the comment about the novelty of this article.

However, considering the interests of the Special Issue (SI) of the Sensors Journal, where the following manuscript is submitted to, we think it is a good fit, since the main purpose of the manuscript is applying AI algorithms within the blockchain technology, whereby one of the SI indicated interests is "Intelligent systems for fraud detection and forensics in blockchain environments". We hope this clarifies the reason why we chose Sensors journal for submitting the manuscript to.

 Summarized changes made to the paper:

An extended English grammar proofreading was performed, which indicated that word – “Anomalous” should be replaced with “Anomaly”, thus we propose a slightly corrected title: “A Machine Learning-based method for automated Blockchain Transaction Signing including Personalized Anomaly Detection” We extended the introduction to explain the obstacles addressed in the cited references. We added a new relevant reference (i.e., 8) published recently in 2019, which proves an actuality of the problem addressed with the described proposed method in the article. We added a description of an application scenario, including a malicious attack scenario, to help readers better understand why we need an anomalous transaction detection system. We changed and improved the description in the contributions and conclusion part of this paper, where we reformulated the previously misused term "single point of failure." With a different color, we have refreshed Figure 3 to highlight the improvements of the proposed method compared to the original protocol. We have rewritten the accompanying text for Figure 2, to be clearer and more precise. We redefined some of the variables and specifically answered some of the questions from the reviewers. We corrected the presentation of the results in the section of the experiment and the discussion of the results in the conclusion. This was done to address the question about the evaluation of the proposed system. Even though any evaluation of such a system is problematic, we specified the reasons for the lack of any scientifically valid evaluation. Furthermore, the description of the usage of such a system was expanded to be clearer that a human is still in the loop of the signing the unusual transactions.

 Reviewer 2 Report

In this paper, the authors propose a new method to enhance the digital signing process in blockchain-related transactions. In particular, their approach involves an automated and personalized digital signing where manual approvals are minimized and are only required by the user in order to avoid anomalous transactions. Additionally, the authors develop a decentralized detection method for anomalous transactions to overcome the bottleneck of single point of failures.

The authors should further elaborate the transaction processing procedure illustrated in Figure 2, e.g., how the feature extraction takes place given the time frame windows and the aggregation functions. The authors should define all variables indicated in Figure 2. Is the rolling window sliding in the transaction history with a fixed length? Is there any method to evaluate the prediction accuracy of the procedure in Figure 2, e.g., is there any prediction confidence interval? In the anomalous transaction detection method (isolation forest), which is the probability of false positive anomalies in the transactions? If it is non-negligible, how this would ultimately affect user's trust on automatic digital signatures?

Other comments/Typos

Page 3, line 99: using user data

Page 6, line 194: with the number as the number of measurements

Page 6, Section 4.2: There is a confusion in the notation of the size of the rolling window.

Page 7, Algorithm 1: There is a typo in the “for loop” for variable w.

Page 11, line 302: address-specific

Author Response

First of all, we would like to thank both the reviewers for the important and helpful remarks, advice, and suggestions concerning the content of the paper. It helped us a lot since we noticed some aspects which had to be improved.

Reviewer #2:

-- In this paper, the authors propose a new method to enhance the digital signing process in blockchain-related transactions. In particular, their approach involves an automated and personalized digital signing where manual approvals are minimized and are only required by the user in order to avoid anomalous transactions. Additionally, the authors develop a decentralized detection method for anomalous transactions to overcome the bottleneck of single point of failures.

 1) The authors should further elaborate the transaction processing procedure illustrated in Figure 2, e.g., how the feature extraction takes place given the time frame windows and the aggregation functions.

Thank you for this comment, we agree that the description of Figure 2 was confusing. The accompanying text describing Figure 2 was rewritten to be clearer.

Changes in the revised version of the manuscript are visible at the following lines: 204-223.

2) The authors should define all variables indicated in Figure 2.

Thank you for this observation. The variables in Figure 2 and the accompanying text were not the same and some of them were not defined. We corrected this and additionally specifically described each one of the variables used in Figure 2.

Changes in the revised version of the manuscript are visible at the following lines: 204-223.

3) Is the rolling window sliding in the transaction history with a fixed length?

Thank you for this question. The answer to this question was added to the rewritten accompanying the description of Figure 2. Citing from the paper:

“Consequently, as the number of transactions varies across the windows, the size of the rolling window is not fixed. The goal of our research is not to determine which of these time frames is most relevant in general, but to build a procedure, which determines the appropriate time frame for each of the addresses individually.”

Changes in the revised version of the manuscript are visible at the following lines: 201-223.

4) Is there any method to evaluate the prediction accuracy of the procedure in Figure 2, e.g., is there any prediction confidence interval?

We appreciate this question. As far as we are aware, there is no scientifically valid metric to determine the quality of unsupervised anomalous transaction detection. For this reason, the charts with the results were shown, to allow the readers to individually determine the effectiveness of the implemented system. It is almost impossible to collect data where transactions are manually labeled as either normal or malicious. This would involve getting in touch with actual cryptocurrency holders and ask them to expose themselves and manually identify the malicious transaction. Also, as we wish for this system to not require the users to manually label their transactions, unsupervised methods are a natural fit.

Changes in the revised version of the manuscript are visible at the following lines: 280-287, 306-309 and 372-379.

 5) In the anomalous transaction detection method (isolation forest), which is the probability of false positive anomalies in the transactions?

Thank you for this question. As the unsupervised anomaly transaction method does not operate with ground-truth labels (ground-truth labels are absent in this kind of problem), there is no scientifically valid way to determine the false positive rate of such methods.

6) If it is non-negligible, how this would ultimately affect user's trust on automatic digital signatures?

We appreciate this question and we thought about this significantly. As there is no way to test the efficacy of this system, users should naturally not completely trust such systems. For this reason, we let the human be in the loop of digitally signing some transaction. The transactions which are following the usual patterns are automatically signed by the system and only unusual transaction prompt the users for its feedback. Of course, malicious transactions that are following the usual pattern would slip by this system. But most common and destructive malicious activities (emptying the wallets) would be prevented with such a system - either a small number of transactions with an unusual amount of currency or an unusual amount of small valued transaction would be caught by the system.

Changes in the revised version of the manuscript are visible at the following lines: 372-379.

7) Other comments/Typos:
Page 3, line 99: using user data.
Page 6, line 194: with the number as the number of measurements.
Page 6, Section 4.2: There is a confusion in the notation of the size of the rolling window.
Page 7, Algorithm 1: There is a typo in the “for loop” for variable w.
Page 11, line 302: address-specific

We appreciate the help with finding the typos. We corrected them.

 Summarized changes made to the paper:

An extended English grammar proofreading was performed, which indicated that word – “Anomalous” should be replaced with “Anomaly”, thus we propose a slightly corrected title: “A Machine Learning-based method for automated Blockchain Transaction Signing including Personalized Anomaly Detection” We extended the introduction to explain the obstacles addressed in the cited references. We added a new relevant reference (i.e., 8) published recently in 2019, which proves an actuality of the problem addressed with the described proposed method in the article. We added a description of an application scenario, including a malicious attack scenario, to help readers better understand why we need an anomalous transaction detection system. We changed and improved the description in the contributions and conclusion part of this paper, where we reformulated the previously misused term "single point of failure." With a different color, we have refreshed Figure 3 to highlight the improvements of the proposed method compared to the original protocol. We have rewritten the accompanying text for Figure 2, to be clearer and more precise. We redefined some of the variables and specifically answered some of the questions from the reviewers. We corrected the presentation of the results in the section of the experiment and the discussion of the results in the conclusion. This was done to address the question about the evaluation of the proposed system. Even though any evaluation of such a system is problematic, we specified the reasons for the lack of any scientifically valid evaluation. Furthermore, the description of the usage of such a system was expanded to be clearer that a human is still in the loop of the signing the unusual transactions.